# Crop classification with UAV multispectral remote sensing, employing an enhanced ResNet50 residual network

Chenwei Xu[1], Shixian Lu[1], Xiang Feng[2], Xuelin Zhang[3], Jianxiong Wang[1]*

1 Yunnan Agricultural University, Yunnan Provincial International Joint Research and Development Center for Smart Environment, Yunnan Agricultural University, Kunming, China, 2 Agricultural Economics and Informatics Research Institute, Yunnan Academy of Agricultural Sciences, Kunming, China, 3 Beijing Century Guoyuan Technology Co., Ltd., Beijing, China

* wangjx@ynau.edu.cn

## Abstract

The ResNet50 residual network model is characterized by its rapid training speed and high classification accuracy, demonstrating significant advantages in extracting and classifying crop features. However, the correlation and redundant information among different spectral bands in multispectral data can negatively affect classification accuracy. Additionally, the ResNet50 model does not meet the requirements for training with multi-band data, and its classification accuracy in small-scale, high-precision scenarios needs further improvement. This study focuses on flue-cured tobacco and maize to address these challenges, identifying the optimal classification band combination. Enhancements to the ResNet50 model were made by incorporating Batch Normalization (BN) layers, pyramid pooling layers, and hidden layers, experimenting with seven different combinations. The experimental results indicate that the RGB＋NIR+Edge combination is the most effective for classifying flue-cured tobacco and maize, achieving an accuracy of 94.48%, precision of 94.66%, recall of 94.48%, kappa coefficient of 91.72%, and an F1 score of 94.49%. Among the seven improvement strategies, solely introducing BN layers yielded the most substantial improvements, increasing accuracy by 0.42%, precision by 0.51%, recall by 0.42%, kappa coefficient by 0.63%, and the F1 score by 0.44%.

## Introduction

Owing to its rapid, extensive, nondestructive, and objective nature, remote sensing technology can effectively identify crop types, planting areas, yields, growth conditions, and disaster information across large regions, enabling precise analysis and offering substantial support for scientific management and decision-making in agricultural production [1]. Crop classification using remote sensing imagery is one of its primary applications [2]. Accurate crop identification and classification provide

**Data availability statement:** All data and code underlying the findings described in this manuscript are fully available without restriction from the Zenodo repository (https://doi.org/10.5281/zenodo.17451761).

**Funding:** This work was supported by the National Key R & D Program (2024YFD1700104). The funders had no role in study design, data collection and analysis, decision to publish, or preparation of the manuscript.

**Competing interests:** The authors declare that they have no conflicts of interest.

insights into the planting structure and farmland distribution, clarify the area and arrangement of various crops, and supply timely and precise data on crop types and growth conditions to agricultural decision-makers, thereby enhancing agricultural productivity [3]. The crop classification results obtained from low- and medium-spatial resolution satellite imagery are insufficient to meet the demands of modern agricultural informatization and precision agriculture [4]. In contrast to conventional satellite remote sensing, UAV-based image acquisition offers higher spatial resolution and captures more detailed crop information, making it particularly suitable for small-scale or high-precision agricultural monitoring. Thus, UAV remote sensing serves as a significant complement to satellite remote sensing in agricultural applications [5].

In this context, UAVs have emerged as optimal instruments for crop classification in agricultural fields owing to their excellent efficiency, cost-effectiveness, and adaptability [6]. Various researchers, both domestically and internationally, have employed UAVs to investigate the detailed classification of major food crops. Li et al. utilized UAV-acquired remote sensing imagery of maize, soybeans, and other crops, combining the ReliefF method with recursive feature elimination based on support vector machines, achieving a classification accuracy of 83.42% [7]. Shuqin Yang et al. proposed a novel approach for classifying UAV multispectral remote sensing images of maize and zucchini by employing the DeepLab V3 + deep semantic segmentation network, enhancing crop classification performance through the extraction of semantic features using an improved model [8]. L Wei introduced a spectral-spatial-position fusion technique that uses conditional random fields for the classification of crops, including peas and soybeans, in hyperspectral UAV remote sensing images. The findings indicate that incorporating spatial context and positional information enhances the use of spatial data, thereby improving the accuracy of crop classification [9]. Feng et al. proposed a recursive convolutional neural network approach that integrates an attention mechanism with multi-temporal UAV imagery. Their results demonstrated that the Bi-LSTM-Attention module outperforms feature stacking and bidirectional LSTM in modeling sequential feature dependencies, while deformable convolution surpasses standard convolution in addressing scale and shape variations within complex agricultural landscapes. Furthermore, the incorporation of multitemporal UAV data enhances the precision of crop fine classification [10].

The sensors currently installed on UAVs mainly consist of digital cameras, multispectral sensors, and hyperspectral sensors. Owing to their reduced cost, digital cameras are limited to collecting wavelengths within the visible spectrum [11]. Although hyperspectral sensors can capture a broader range of wavelength bands, their high cost, complex data processing requirements, and significant technological barriers collectively limit their widespread application in agriculture [12]. In contrast, multispectral sensors are more commonly used in agricultural remote sensing due to their affordability and their ability to cover the red-edge and near-infrared bands, which are critical for monitoring vegetation growth [13]. Luo, S using a multispectral UAV to observe several rice cultivars across many locations. The segmented empirical line method was proposed to enhance the accuracy of reflectance acquisition from multispectral remote sensing data by comparing the effects of various

radiometric calibration methods and UAV flight altitudes. Additionally, the vegetation index was employed to estimate the growth parameters and yield of rice. The findings indicate that the segmented empirical line approach outperforms alternative methods regarding estimation accuracy and exhibits reduced impact on outcomes when the flying altitude exceeds 100 meters [14]. Tao, C employs ResNet-50 to extract features from multispectral remote sensing images and integrates a feature pyramid architecture to amalgamate multi-scale features for effective semantic segmentation. Experiments demonstrate that MSNet exhibits strong performance on the WHU GID and ISPRS Potsdam datasets, particularly in the 6c data mode, where the recognition accuracy of vegetation and artificial features is markedly enhanced [15]. Denis, A offers an approach utilizing multispectral remote sensing to facilitate organic crop certification, thereby mitigating the challenges and expenses associated with field inspections by differentiating between organic and conventional cornfields. The findings indicate that multispectral satellites attain full or nearly full separation and proficiently distinguish between organic and conventional cornfields by employing high spatial resolution and targeted spectral bands (e.g., 550 nm for green light, 700 nm for red edge, and 725 nm for near-infrared), along with pertinent vegetation indices (e.g., Chlorophyll Content Index (CCI), Normalized Vegetation Index (NDVI), and Greenshape Index (G)) [16].

Currently, UAV multispectral image crop fine categorization encounters problems. Multispectral data consist of numerous bands with complex spectral features and substantial dynamic variations, where the correlation and redundancy among different bands complicate feature extraction and adversely affect classification accuracy [17]. Additionally, the ResNet50 model continues to exhibit adaptation challenges and requires improvements in classification accuracy when processing multispectral data, while also facing significant difficulties in balancing model complexity with performance.

In this study, the ResNet-50 architecture was modified to improve the mapping of deep features to the final classifier, thereby enhancing the classification performance of multispectral remote sensing images. This study initiated data pretreatment and input optimization, rigorously identifying the ideal parameter configuration for multispectral remote sensing classification tasks using rigorous ablation experiments. The experimental findings demonstrate that a patch size of 57 × 57 pixels is ideal while also confirming the substantial benefit of the RGBEN band combination regarding information discrimination capabilities, thereby providing a solid data basis for effective model learning. In this study, we carefully examined the design of a tail feature augmentation module for model structure optimization. Three fundamental modules were developed: a spatial feature refinement layer, nonlinear feature projection layer, and feature vector distribution normalization layer. A collection of feature aggregation optimization algorithms specifically designed for the ResNet architecture was developed through a thorough assessment of seven combination approaches. An ideal performance improvement structure was identified through comprehensive experimental comparisons and mechanistic analysis. This study examined how various optimization modules improve model generalization and classification accuracy through feature discriminative power and distribution stability. This offers systematic design strategies and theoretical insights for enhancing deep learning architectures in remote sensing.

## Materials and methods

### Materials

The research site is located in Youshuo Town, Chengjiang City, Yuxi City, Yunnan Province. Chengjiang City (24°29'–24°55'N latitude and 102°47'–103°04'E longitude) experiences a central subtropical plateau monsoon climate, characterized by moderate temperatures throughout the year, abundant sunlight, and the concurrence of rainfall and warmth within the same season [18].

The primary crops in the study area are tobacco and maize at their respective maturity stages; maize is cultivated in the red-boxed area, while tobacco is grown in the green-boxed area. An overview of the research region is shown in Fig 1.

### Data collection

The experimental data were collected under overcast conditions on August 19 and August 27, 2024, using the DJI Elf 4 multispectral drone. To ensure data precision and reliability, DJI GS Pro was employed for route planning, incorporating

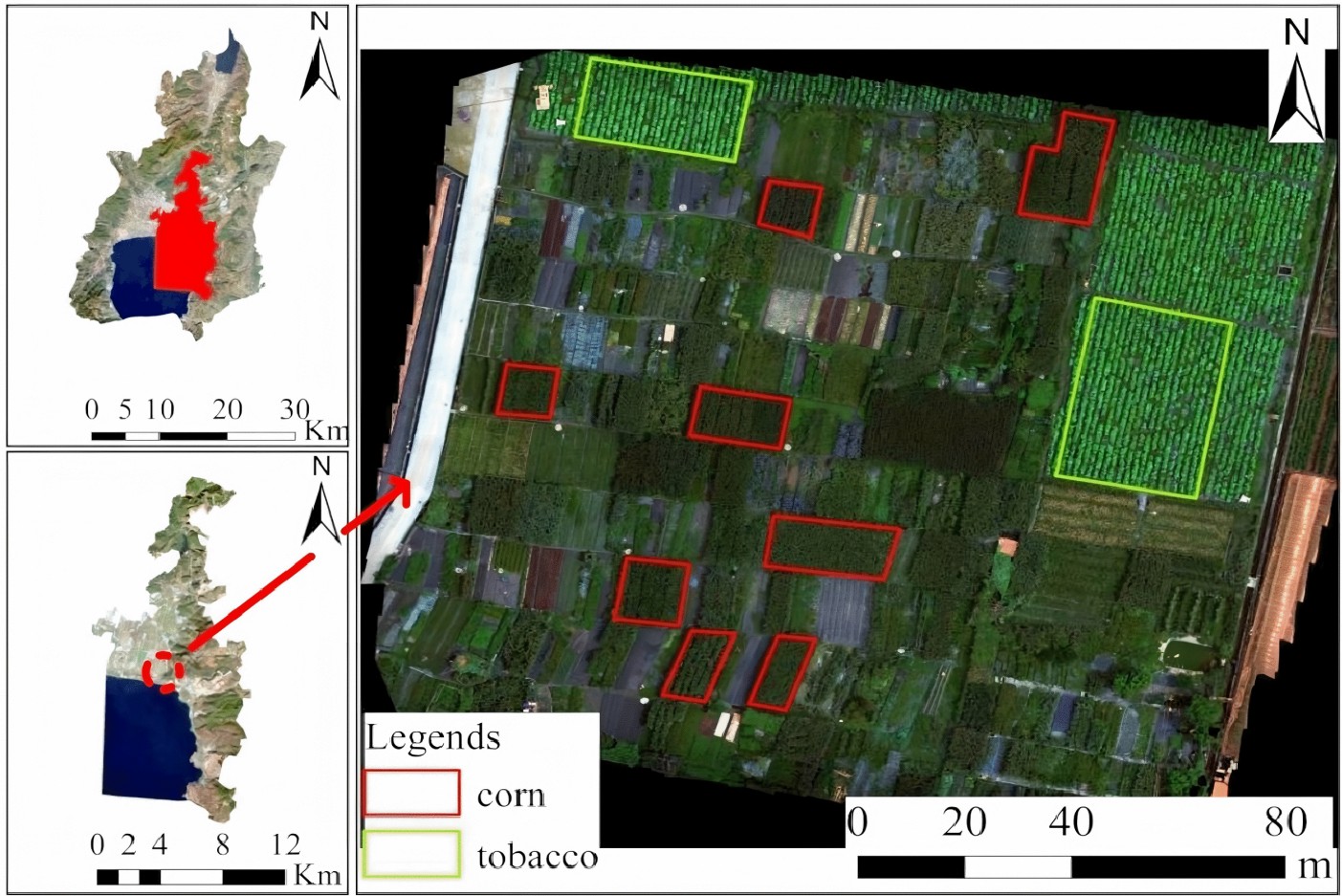

**Fig 1. Study area.**

406 waypoints, 28 primary routes, a total route length of 3,114 m, a heading overlap rate of 62%, a bypass overlap rate of 67%, a central route angle of 70°, a flight altitude of 25 m, a ground sample distance of 1.3 cm/pixel, and a photo mode involving hovering at waypoints perpendicular to the main route. The picture mode involves a waypoint hovering with the photographic orientation perpendicular to the primary flight trajectory. Ultimately, 2060 TIFF files were obtained.

## Data preprocessing

Radiometric Calibration and Image Mosaicking: The DJI Wizard P4M is equipped with a sunlight sensor that automatically records lighting conditions. PIX4Dmapper and ENVI 5.6 were employed for radiometric and geometric correction, ultimately stitching the acquired images into orthophotos. The resulting orthophotos consist of five bands: red, green, blue (RGB), red edge (Edge), and near-infrared (NIR), with all data types formatted as 32-bit floating-point. The projected coordinate system for all image data in this study is WGS 1984 (World Geodetic System 1984), a global geographic coordinate system.

Dataset Generation. Label production was conducted via ENVI Classic 5.6 and LABELME, resulting in three labels: 'tobacco', 'corn', and 'background', as illustrated in Fig 2. Following labeling, a JSON file documenting the coordinates of each labeled box is exported. The dataset is then organized and classified according to the labeling type. Subsequently, the dataset is divided into a training set and a test set at an 8:2 ratio.

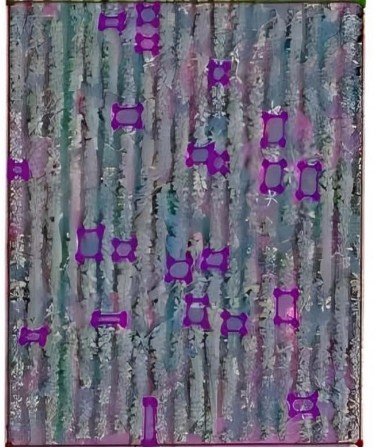 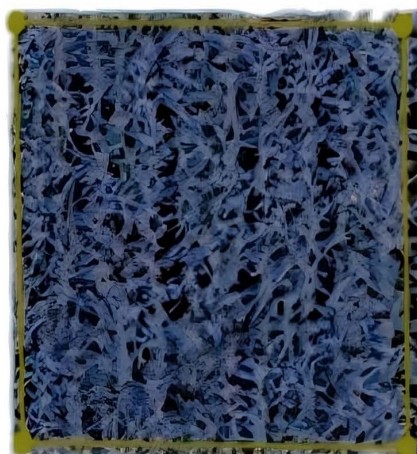 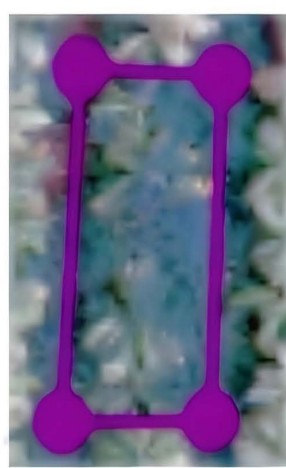

**Fig 2. Exemplary labels.**

## Methods

**Modeling of the ResNet50 residual network.** ResNet50 is a classical deep convolutional neural network widely employed in image classification and various other visual tasks. It addresses the problems of gradient vanishing and gradient explosion in deep networks through the use of residual connections. The model consists of multiple convolutional layers, batch normalization layers, activation functions, pooling layers, and fully connected layers [19,20]. Let the model comprise $L$ layers, with the output of each layer represented as $X^l$ ($l = 1, 2, ...L$). The transformation of each layer can be expressed as: $X^l = f^l (X^{l-1}; W^l)$. The alteration of a concrete structure's layer is referred to as:

$$X^1 = f^1 \left( X^0; W^1 \right) = RELU \left( BatchNorm \left( Conv \left( X^0; W^1 \right) \right) \right) \tag{1}$$

$$X^2 = f^2 \left( X^1; W^2 \right) = RELU \left( BatchNorm \left( Conv \left( X^1; W^2 \right) \right) \right)$$
$$\vdots \tag{2}$$

$$X^L = f^L \left( X^{L-1}; W^L \right) = RELU \left( BatchNorm \left( Conv \left( X^{L-1}; W^L \right) \right) \right) \tag{3}$$

In layer $l$, $f^l$ () denotes the transformation function, encompassing the convolutional layer, ReLU nonlinear activation function, pooling layer, etc., while $W^l$ represents the weight parameter inside layer l. The input image is denoted as $X^0$. The *BatchNorm* () function represents the batch normalization layer, which mitigates internal covariate shift and addresses issues of gradient vanishing and gradient exploding, hence enhancing the stability and speed of the model during training [21].

ResNet50 is employed as the foundational model and is suitably adapted to accommodate multispectral image inputs. The modifications are as follows: the convolutional kernel in the input layer is expanded from 3 to 5 channels to align with the number of bands in the multispectral images. A dropout layer, with a dropout rate of 0.5, is introduced before the fully connected layer to mitigate overfitting. To enhance the model's recognition capability for multiscale objects, a pyramid pooling layer is integrated between the final convolutional layer and the global pooling layer, utilizing pooling dimensions of 1×1, 2×2, 3×3, and 6×6. Additionally, hidden and normalization layers are incorporated before the fully connected

layer following global pooling to accelerate the training process, improve model stability, and strengthen feature extraction and representation capabilities. The outcomes of the enhanced model are illustrated in Fig 3.

**Pyramid pooling layer (PPL).** The pyramid pooling layer captures both global and local contextual information through global average pooling operations at multiple scales, subsequently integrating this information into the feature map [22,23]. The pyramid pooling layer in this study consisted of several concurrent pooling branches, each executing global average pooling operations of varying sizes on the input feature maps. Each resulting pooled feature map is then processed by a 1 × 1 convolutional layer, reducing the number of channels to one-fourth of the original input to enhance computational efficiency and facilitate feature integration. Following this, the reduced feature maps are upsampled via bilinear interpolation to match the spatial dimensions of the original feature map. Finally, all feature maps are concatenated along the channel dimension and subjected to a nonlinear transformation using the ReLU activation function. During the forward propagation of the model, the PPL is implemented subsequent to the output of the last convolutional layer:

$$x = PPL(x) \tag{4}$$

PPL() signifies the processing of the input feature map x by the pyramid pooling layer. The procedure is illustrated in Fig 4 Pyramid pooling layer.

PPL improves the model's capacity to identify multiscale targets. By aggregating operations across many dimensions, multiscale properties ranging from global to local are acquired, hence enhancing the model's robustness to targets of differing scales [24].

**Hidden layers.** The hidden layer enhances the model's expressive capacity, facilitates the acquisition of more intricate feature representations, and further advances feature extraction and integration. The nonlinear activation function (ReLU)

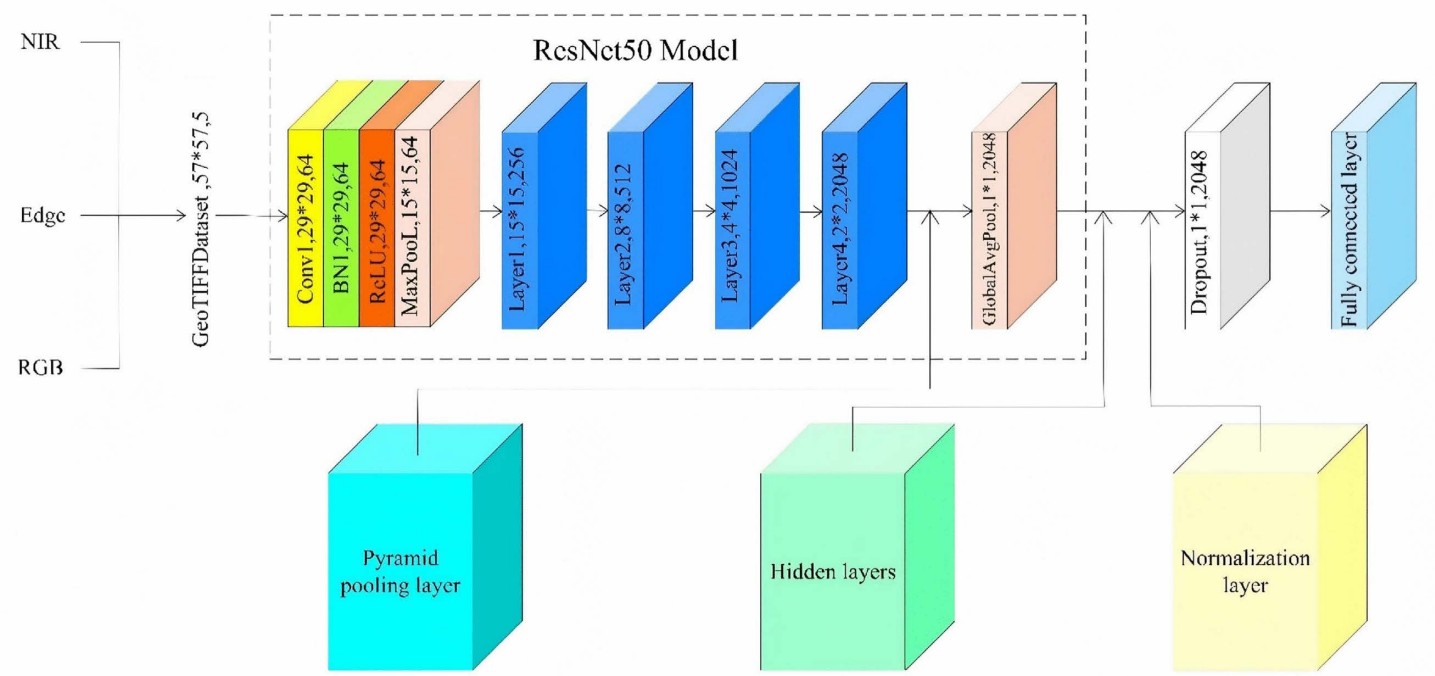

**Fig 3. Improved ResNet50 model structure.**

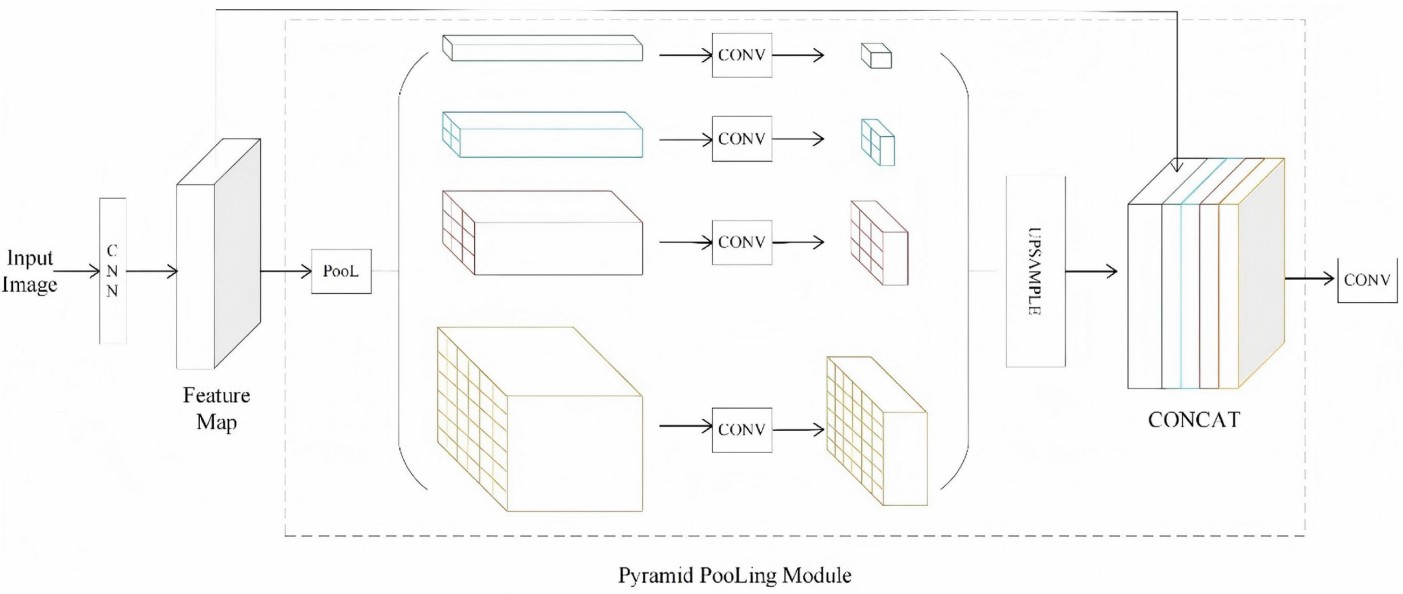

**Fig 4. Pyramid pooling layer.**

employed within the hidden layer improves the model's ability to capture nonlinear relationships, thereby increasing its effectiveness in addressing complex classification problems [25,26]. In the forward propagation of the enhanced experiment detailed herein, the hidden layer acquired more intricate coupling relationships across the dimensions of the global feature vector by introducing further nonlinear transformations. This mitigates the potential information loss resulting from the global average pooling procedure. Subsequently to the GAP procedure, the deep feature maps were condensed into a 1×C global feature vector $\chi_{GAP}$. Despite $\chi_{GAP}$'s global perspective, information within its dimensions is frequently linearly coupled and lacks adequate discriminative capability. In multispectral remote sensing classification tasks, the significant correlation among several spectral bands necessitates models with enhanced feature-decoupling abilities. The HID layer transforms $\chi_{GAP}$ into a novel feature space using a fully linked layer and a ReLU nonlinear activation function, thereby augmenting the expressive capacity of the model. This nonlinear reconstruction enhances feature extraction and fusion, thereby improving the adaption to intricate classification problems. The functionality of the HID layer can be articulated as

$$\chi_{HID} = ReLU\left(W_{f_c} \cdot \chi_{in} + b\right) \tag{5}$$

where $\chi_{in}$ represents the input feature vector of the HID layer, $W_{f_c}$ denotes the weight matrix, $b$ denotes the bias vector acquired from the fully connected layer, and $ReLU(\cdot)$ is the nonlinear activation function applied to each element of the feature vector. $\chi_{HID}$, as the anticipated characteristic, was subsequently input into the BN layer for distribution normalization to guarantee its stability.

**Batch normalization (BN) layer.** The BN layer standardizes the input data to achieve a mean of 0 and a standard deviation of 1, hence mitigating internal covariate shifts [27]. Additionally, the BN layer introduces a regularization effect that reduces the risk of overfitting and enhances the model's generalization performance [28]. The intricate nature of multispectral image data and nuanced distinctions within categories render deep feature vectors prone to instability or drift following sophisticated operations, including global average pooling layers and hidden layers. This instability obstructs

learning in the final classification layer, particularly in remote sensing classification contexts, where dataset heterogeneity intensifies the problem. Consequently, we designated the BN layer as a generalization enhancement head subsequent to the hidden layer to guarantee that the features provided to the final classifier demonstrated a highly stable distribution. The mathematical functions of the BN layer were delineated using the following two equations:

$$\hat{\chi}_{in} = \frac{\chi_{in} - \mu_B}{\sqrt{\sigma_B + \varepsilon}}$$

(6)

$$y_{out} = \gamma \odot \hat{\chi}_{in} + \beta$$

(7)

Here, $\mu_B$ and $\sigma_B$ represent the mean and standard deviation of the feature vector, respectively, dynamically calculated from the current mini-batch training data; $\hat{\chi}_{in}$ denotes the normalized vector with zero mean and unit variance; and $\varepsilon$ is a minimal constant employed to ensure numerical stability. The ultimate output of the BN layer is $y_{out}$. The incorporation of the learnable scaling factor $\gamma$ and bias factor $\beta$ allows the network to maintain a stable feature distribution, while retaining its nonlinear expressive capacity. A stabilizing approach for deep discriminative features is essential for improving the robustness and generalization performance of intricate multispectral remote sensing data.

## Experiments and schemes

**Comparative analysis of multi-scale input strategies.** The input patch size is a critical parameter that influences the receptive field and feature granularity of the model. To systematically evaluate the information-gathering capabilities across several spatial scales and provide a benchmark that reconciles computational efficiency with contextual information for this study, four patch sizes were formulated for comparison. This method analyzes the impact of the input scale on the efficacy of the classification models by adjusting the variables.

**Band ablation experiment.** Multispectral imagery consists of multiple bands, each providing distinct spectral information; however, not all bands contribute equally to crop classification, and different band combinations can significantly influence feature extraction and classification accuracy. Therefore, the classification effects of various band combinations are systematically evaluated through ablation experiments to establish an optimal dataset for future model improvements. In this study, four distinct waveband combinations are used as model inputs, as presented in Table 1 of the ablation experiments, and the optimal combinations are identified by calculating the evaluation metrics corresponding to each combination.

**Baseline model selection experiment.** This study conducted a thorough model comparison analysis based on the identification of optimal data input configurations, focusing on the fundamental properties of multispectral remote-sensing scene classification tasks. This study sought to rigorously examine the performance discrepancies among several deep learning models in this task, arising from their intrinsic inductive biases and structural priors. The chosen candidate models comprised ResNet-50, DenseNet-121, Darknet-53, and EfficientNet-B0, exemplifying essential structural concepts, including residual learning, feature reuse, efficient backbone architecture, and composite dimension scaling. The assessment focused on two fundamental aspects: the number of model parameters and the classification precision.

**Table 1. Ablation experiments.**

| Ablation experiment serial number | Swing portfolio |
| --- | --- |
| 1 | RGB |
| 2 | RGB + NIR |
| 3 | RGB+Edge |
| 4 | RGB + NIR+Edge |

This study sought to quantitatively identify architectures that optimize the balance between representational capacity and resource efficiency, thereby creating a robust benchmark platform for future structural innovations aimed at improving the discriminative power and generalization abilities of models.

**Enhanced experiments utilizing ResNet-50.** This study develops seven distinct combinations of enhancement algorithms to comprehensively evaluate their effects on UAV multispectral image-based crop classification performance, balancing model complexity with efficacy. Through a systematic comparison of the seven enhancement schemes detailed in Table 2, the impact of each strategy on model performance improvement is analyzed, concurrently revealing synergistic effects or potential conflicts among various strategies. This analysis establishes a scientific basis for future model optimization, ensuring appropriate model structure and optimization strategy selection in complex application scenarios, thus preventing unnecessary increases in model complexity while enhancing classification accuracy.

The placement of the batch normalization (BN) layer, responsible for data normalization, before convolutional, fully connected, or hidden layers may result in unstable weight updates, gradient vanishing or explosion, and activation function saturation, thereby negatively affecting model training and performance. Accordingly, the optimized BN layer placement is after these layers, as shown in Table 3.

Importantly, the BatchNorm layer following the pyramid pooling layer is BatchNorm2d, whereas all other layers utilize BatchNorm1d.

**Training curriculum.** The GeoTIFFDataset class is intended for processing remote sensing imagery in the GeoTIFF format. To address the imbalance in picture amounts across categories, we developed a systematic sampling approach to ensure uniformity in the number of samples for each category throughout model training. We initially implemented a category-based balanced sampling technique at the data-sampling level. This approach preserves the quantitative balance among categories during data loading by ensuring uniform sample sizes for each category.Second, in the design of the loss function, we incorporated Focal Loss and adjusted category weight values. By allocating suitable weight coefficients to various categories, we improved the sensitivity of the model to sample imbalance, thus alleviating the

**Table 2. Improvement of the experimental program.**

| Improvement experiment serial number | Programmatic |
|---|---|
| 1 | PPL |
| 2 | BN layer |
| 3 | hidden layer |
| 4 | PPL+BN layer |
| 5 | PPL+hidden layer |
| 6 | BN layer+hidden layer |
| 7 | PPL+BN layer+hidden layer |

**Table 3. BN layer locations.**

| Combinatorial approach | Location |
|---|---|
| ResNet50+BN | Behind the convolutional layer |
| ResNet50+BN+hidden layer | Behind the Hidden Layer |
| ResNet50+BN+PPL | Pyramid Pooling Layers following Convolutional Layers |
| ResNet50+BN+hidden layer+PPL | Pyramid pooling layer beyond the convolutional layer, beyond the hidden layer |

training bias resulting from the disproportionate distribution of easy and difficult samples among categories. Furthermore, throughout the data-loading procedure, we utilized a unique balanced sampler to create a data loader. This sampler rigorously upholds equitable sample counts across categories at the batch level, guaranteeing uniform occurrence frequencies for each class throughout each training batch. This method inhibits the model by forming category biases during the training.

Furthermore, data augmentation techniques, including random horizontal and vertical flipping, rotation (±15°), Gaussian blurring (kernel size of 3, standard deviation between 0.1 and 2.0), and normalization, are applied. Normalization scales the pixel values in each band to the range [−1, 1], with a mean and standard deviation of 0.5. Model training is conducted using the Adam optimizer, with a learning rate of $1 \times 10^{-5}$ and a weight decay of $1 \times 10^{-5}$. The cross-entropy loss function is utilized for optimization.To evaluate the reliability of the trial outcomes and minimize variability, four independent experiments were performed for each model using four random seed values (42, 52, 62, and 72).The DataLoader in PyTorch was employed to load training and validation datasets with a batch size of 64; after each epoch, the model performs forward propagation, computes the loss, and then adjusts the parameters via backpropagation. Upon completion of each training cycle, the model evaluates the validation set and measures accuracy, precision, recall, Kappa coefficient, and F1 score, utilizing the early stopping technique to regulate the training process and prevent overfitting. The early stopping strategy involves terminating training when the validation set accuracy does not improve after a predefined number of training cycles, thereby mitigating the risk of overfitting due to extended training duration [29].

**Assessment metrics.** The evaluation index uses accuracy, which is the ratio of the number of samples correctly classified by the model to the total number of samples; the closer the accuracy rate is to 1, the greater the effectiveness of the classifier in classifying the overall data. Precision refers to the proportion of samples predicted by the model to be positive classes that are, in fact, positive classes. A precision rate closer to 1 indicates a higher proportion of correctly predicted positive samples, signifying greater reliability of the model's predictions. Recall, on the other hand, refers to the ratio of samples correctly identified as positive by the model to the total number of actual positive samples. A recall rate approaching 1 indicates superior model performance in identifying true positive class samples. The kappa coefficient (Cohen's kappa) quantifies the classification accuracy; a kappa value approaching 1 indicates superior classifier performance relative to random classification. The F1 Score is the harmonic mean of precision and recall, providing a comprehensive measure of the model's classification performance. An F1 Score approaching 1 indicates stronger precision and recall, reflecting enhanced performance in the identification and predictive reliability of positive classes.

## Results

### Comparative experimental outcomes of multi-scale input strategies

This study utilized ResNet-50 as the foundational architecture to examine the effect of input size on model performance by employing comprehensive information as input. We methodically assessed the impact of four different input patch dimensions ($37 \times 37$, $47 \times 47$, $57 \times 57$, and $67 \times 67$) on model performance. Table 4 shows the performance characteristics of the model for the validation set across various patch sizes.

The data in Table 4 indicate that when the input patch size increased from $37 \times 37$ pixels to $57 \times 57$ pixels, the classification performance of the ResNet-50 model was enhanced across all metrics. At $57 \times 57$ dimensions, the model attained peak performance, with validation accuracy, F1 score, and Kappa coefficient of 94.48%, 94.49%, and 91.72%, respectively. Upon increasing the size to $67 \times 67$, the model performance deteriorated, resulting in an accuracy reduction of 94.01%. This suggests that although larger patch sizes offer increased spatial information, they may also introduce redundant backdrops, thereby hindering feature discrimination. Considering performance and efficiency, a dimension of $57 \times 57$ achieves ideal equilibrium between spatial information acquisition and model expressiveness. Consequently, this was designated as the standardized input size for the ensuing studies.

**Table 4.  Performance Comparison of ResNet-50 with Varied Patch Dimensions.**

| patch size | Validation Accuracy | Precision | Recall | Kappa | F1 Score |
|---|---|---|---|---|---|
| 37 | 90.17% | 90.49% | 90.17% | 85.25% | 90.23% |
| 47 | 92.19% | 92.34% | 92.19% | 88.28% | 92.22% |
| 57 | 94.48% | 94.66% | 94.48% | 91.72% | 94.49% |
| 67 | 94.01% | 94.11% | 94.01% | 91.02% | 94.02% |

## Results of band ablation experiments

Based on the determination that 57×57 is the optimal patch size, this section further systematically evaluates the impact of input band combinations on the model's classification performance. Comparing four band combination schemes, Table 5 presents the model performance under different band combinations.

The outcomes of the band ablation experiments unequivocally demonstrated that the full-band combination achieved superior classification performance across all evaluation metrics, achieving a validation set accuracy of 94.48% and a kappa coefficient of 91.72%, thereby illustrating the synergistic advantage of incorporating multispectral information. Initially, the incorporation of NIR and Edge bands, in contrast to RGB, resulted in consistent performance enhancements, with accuracy increases between 0.4% and 0.77%. This confirms the essential function of nonvisible bands in the classification of multispectral remote sensing images. Significantly, the incorporation of the red-edge band resulted in superior performance compared with the inclusion of the near-infrared band.

To investigate the mechanism underlying the enhanced performance of the full-band combination, we randomly sampled 1,000 points and examined the spectral distribution features of the vegetation categories, as illustrated in Fig 5.

Fig 5a and 5b illustrate the spectral feature distributions of tobacco and maize in the NDVI, near-infrared, and red-edge bands, respectively, revealing notable disparities in their spectral properties. In terms of the distribution patterns, maize (Fig 5b) exhibited typical high-value clustering across all three spectral dimensions. The NDVI values primarily aggregated above 0.6, although the near-infrared and red-edge bands also sustained high numerical values. Conversely, tobacco (Fig 5a) exhibited notably different spectral distribution patterns: values across all bands dramatically shifted toward lower ranges, accompanied by broad distribution spans. Specifically, in the NDVI dimension, tobacco values primarily remained below 0.6, thus establishing a distinct demarcation from maize.Disparities in the red-edge bands were very prominent. Corn demonstrated a pronounced rapid ascent in the red-edge zone, whereas tobacco showed a smooth distribution. This distinction offers essential spectral evidence to precisely distinguish between the two crop types. This high spectral separability is of considerable practical significance from a remote-sensing classification standpoint. Non-visible bands significantly improve the discriminative capacity of the feature space by identifying inherent variations in crop physiological structures and biochemical compositions, thus enabling deep learning models to derive discriminative features with distinct biophysical relevance. The synergistic effect of multisource spectral data allows models to use these inherent spectral variations more comprehensively. As a result, full-band combinations provide more accurate crop identification in multispectral remote-sensing categorization.

**Table 5.  Results of ablation test.**

| Swing portfolio | Validation Accuracy | Precision | Recall | Kappa | F1 |
|---|---|---|---|---|---|
| RGB | 93.55 | 93.82 | 93.55 | 90.32 | 93.57 |
| RGB+NIR | 93.95 | 94.10 | 93.95 | 90.92 | 93.96 |
| RGB+Edge | 94.32 | 94.44 | 94.32 | 91.48 | 94.33 |
| RGB+NIR+Edge | 94.48 | 94.66 | 94.48 | 91.72 | 94.49 |

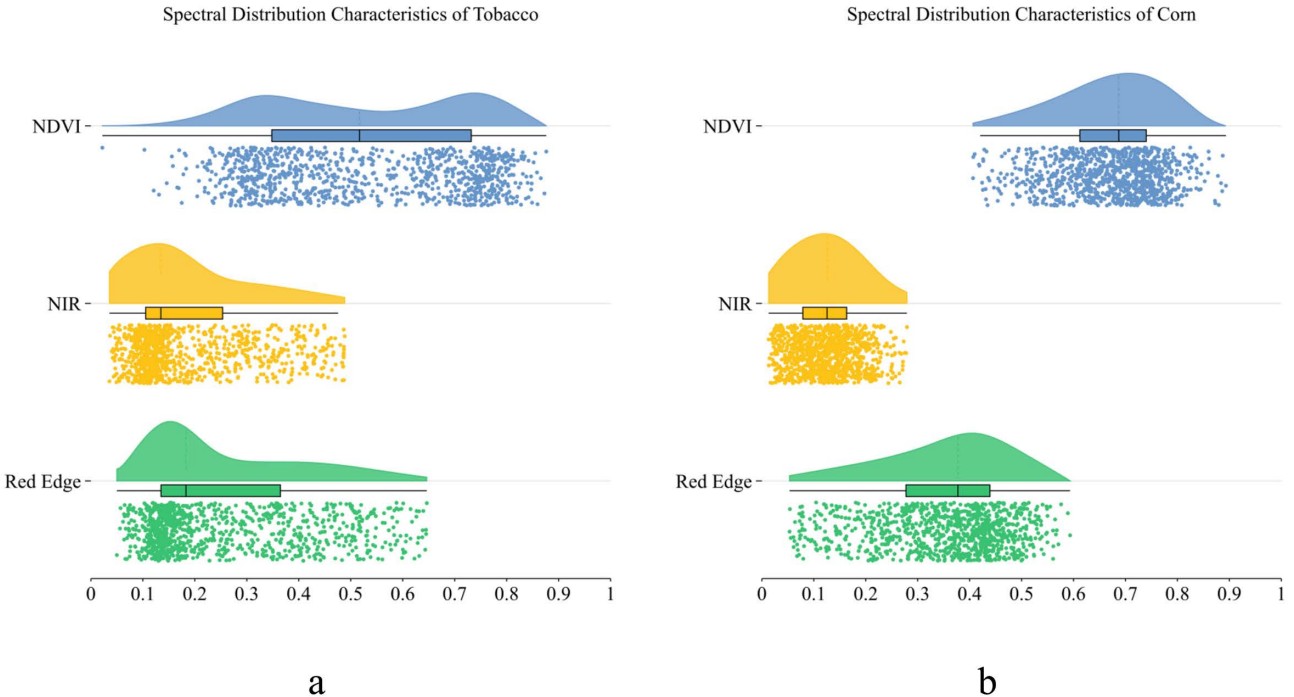

Fig 5. **Differential Spectral Information Distribution Between Tobacco and Corn.**

This study identifies the 57×57 full-band combination as the ideal input configuration for all subsequent model comparisons and enhancement studies based on the quantitative results and spectral mechanism analysis presented above.

## Comparative experiment with baseline models

This section evaluates the classification performance, efficiency, and stability of four prominent CNN architectures: ResNet-50, DenseNet-121, Darknet-53, and EfficientNet-B0, as illustrated in this paper. The objective was to substantiate the selection of ResNet-50 as the baseline for feature-aggregation enhancement networks. All the models utilize optimal input configurations. Table 6 shows the classification performance on a single test set and the model complexity characteristics for each architecture.

ResNet-50 exhibited exceptional classification performance, attaining a validation set accuracy of 94.48%. The F1 score of 94.49% and Kappa coefficient of 91.72% exceeded those of the other baseline models. This exceptional performance arose from the design benefits of the residual-connection mechanism. This approach establishes cross-layer feature reuse paths, significantly alleviating the vanishing gradient problem in deep networks and facilitating deep integration

Table 6. **Model Performance Comparison.**

| Model | Validation Accuracy | Precision | Recall | Kappa | F1 Score | Parameter Quantity | flops |
|---|---|---|---|---|---|---|---|
| ResNet-50 | 94.48 | 94.66 | 94.48 | 91.72 | 94.49 | 23.520M | 332.986M |
| DarkNet-53 | 93.95 | 94.13 | 93.95 | 90.92 | 93.97 | 40.588M | 569.725M |
| EfficientNet-B0 | 92.44 | 92.60 | 92.44 | 88.66 | 92.43 | 4.011M | 33.415M |
| DenseNet-121 | 89.52 | 90.10 | 89.52 | 84.28 | 89.43 | 6.963M | 182.041M |

of spectral and spatial data across several layers. Thus, it markedly improved the capacity of the model to identify multi-spectral remote sensing imagery.

Notable disparities arose among the architectures regarding model complexity. Despite DarkNet-53 attaining the second-highest classification accuracy at 93.95%, its model parameters (40.59 million) and computational cost (569.73 million FLOPs) were significantly higher than those of the alternative models, suggesting the potential for efficiency optimization. Conversely, EfficientNet-B0, an exemplar of lightweight architecture, demonstrates superior model efficiency with 4.01 million parameters and 33.42 million FLOPs. Nonetheless, its classification accuracy of 92.44% significantly lagged behind that of the optimal model, indicating an intrinsic trade-off between accuracy and efficiency. DenseNet-121 exhibited suboptimal performance, attaining an accuracy of merely 89.52%. This indicates that its dense connection mechanism may experience architectural deficiencies while processing remote sensing data with intricate spectral attributes, particularly when confronting issues related to feature redundancy management and efficient feature selection.

To assess the resilience and stability of each baseline model throughout the training period, we generated box plots derived from the F1 Score findings of these four independent experiments. Fig 6 illustrates the notable disparities across architectures in terms of training stability. Among the baseline models in this investigation, ResNet-50 exhibited superior resilience, characterized by a remarkably narrow box range focused inside the high-value interval, with a median of approximately 94.5%. This signifies that the model is impervious to training randomness and possesses remarkable generalization ability. Despite DarkNet-53 exhibiting commendable consistency, its median performance and distribution range were subpar compared to those of ResNet-50. Conversely, EfficientNet-B0 and DenseNet-121 demonstrate significant performance variability. DenseNet-121 exhibited the broadest box span and statistical outliers with a minimum value of 85.0%, indicating its heightened sensitivity to parameter initialization and lack of stability. Despite EfficientNet-B0 attaining a satisfactory performance, its substantial interquartile range signifies intrinsic uncertainty in the training procedure. In conclusion, ResNet-50 functioned as an adequate benchmark. It not only achieves superior classification performance, but also exhibits remarkable training robustness, establishing a dependable basis for future model enhancements.

## Empirical findings on model enhancement

Table 7 presents the results of the model enhancement studies. Fig 7 illustrates the performance progression patterns and convergence attributes of the eight deep learning models throughout the training phase.

This study revealed that the kappa coefficient of all models surpassed 90%, demonstrating a significant level of concordance between the model classifications and the actual labels through the comparison of performance metrics across various improvement approaches. The model that included a Batch Normalization (BN) layer on top of the RGB+NIR+Edge feature set exhibited superior performance in terms of accuracy, Rrecision, Recall, Kappa coefficient, and F1 score. The model attained an Accuracy of 94.90%, Precision of 95.17%, Recall of 94.90%, Kappa coefficient of 92.35%, and an F1 score of 94.93%. The performance remains consistent as the number of training iterations increases. Fig 8 shows the classification results of the ResNet50_BN model.The combination of RGB, NIR, Edge, PPL, and HID exhibited the lowest categorization accuracy. Following its implementation, the Accuracy diminished by 0.49%, Precision fell by 0.54%, Recall reduced by 0.49%, and F1 score plummeted by 0.49%. All six enhancement strategies, with the exception of the RGB+NIR+Edge+PPL+HID combination, improved classification accuracy. Despite the RGB+NIR+Edge+BN+PPL+HID combination attaining the highest F1-SD score of 0.00038, signifying exceptionally stable training, its overall performance was marginally inferior to that of previously described ideal models. This indicates that the model design requires a balance between stability and optimal performance.

The validation accuracy curves indicate that all models demonstrate a fast increase during the initial training phase, eventually stabilizing after approximately the 10th training cycle. This signifies a robust model convergence. The RGB+NIR+Edge+BN model had the highest validation accuracy of 94.90%, characterized by a consistent training

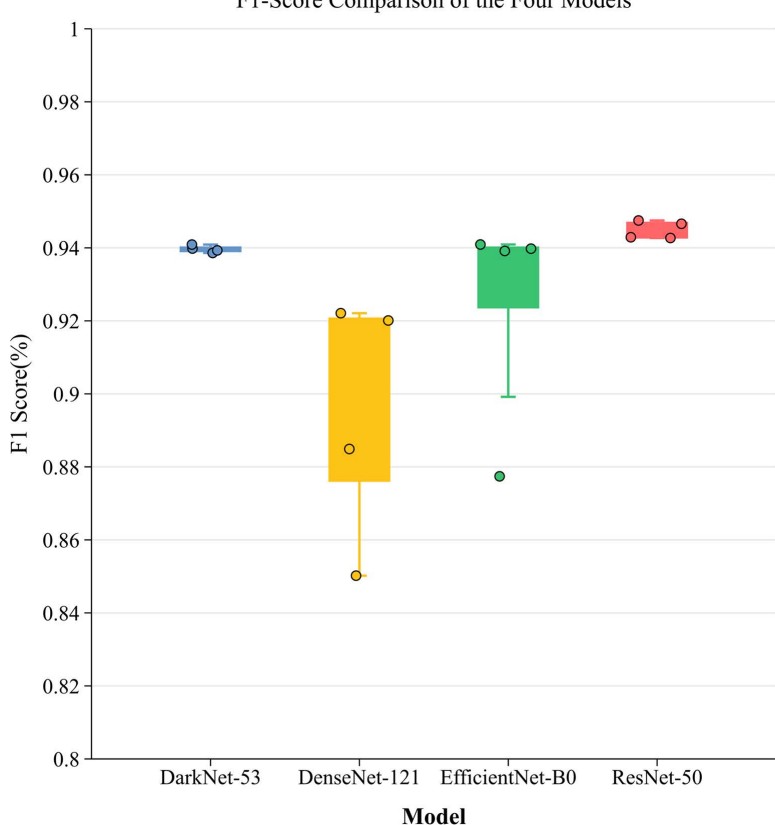

**Fig 6. Comparative F1-Score Distributions Across Models.**

**Table 7. Enhanced Experimental Outcomes.**

| Combination | Validation Accuracy | Precision | Recall | Kappa | F1 | F1-SD |
|---|---|---|---|---|---|---|
| RGB + NIR+Edge | 94.48 | 94.66 | 94.48 | 91.72 | 94.49 | 0.00248 |
| RGB + NIR+Edge+BN | 94.9 | 95.17 | 94.9 | 92.35 | 94.93 | 0.00243 |
| RGB + NIR+Edge+HID | 94.87 | 95.14 | 94.88 | 92.31 | 94.9 | 0.00175 |
| RGB + NIR+Edge+PPL + BN | 94.85 | 95.16 | 94.85 | 92.28 | 94.88 | 0.00233 |
| RGB + NIR+Edge+HID + BN | 94.8 | 95.1 | 94.8 | 92.2 | 94.83 | 0.00150 |
| RGB + NIR+Edge+PPL | 94.74 | 95.06 | 94.75 | 92.11 | 94.78 | 0.00161 |
| RGB + NIR+Edge+BN + PPL + HID | 94.72 | 95.02 | 94.72 | 92.08 | 94.75 | 0.00038 |
| RGB + NIR+Edge+PPL + HID | 93.99 | 94.12 | 93.99 | 90.98 | 94 | 1.01983 |

HID denotes the hidden layer, BN represents the Batch Normalization layer, and PPL symbolizes the Pyramid Pooling layer. F1-SD represents the variability in the sample annotation over the four F1 scores generated from different random seeds.

process and low variations. The RGB + NIR+Edge+HID and RGB + NIR+Edge+PPL + BN models attained accuracies of 94.87% and 94.85%, respectively. The comparable performance of all three models suggests that the BN layer and hidden layers enhance model efficacy. Conversely, the RGB + NIR+Edge+PPL + HID model exhibited subpar performance with a validation accuracy of 93.99%. The standard deviation of the F1 score (F1-SD) was 1.01983, which was markedly

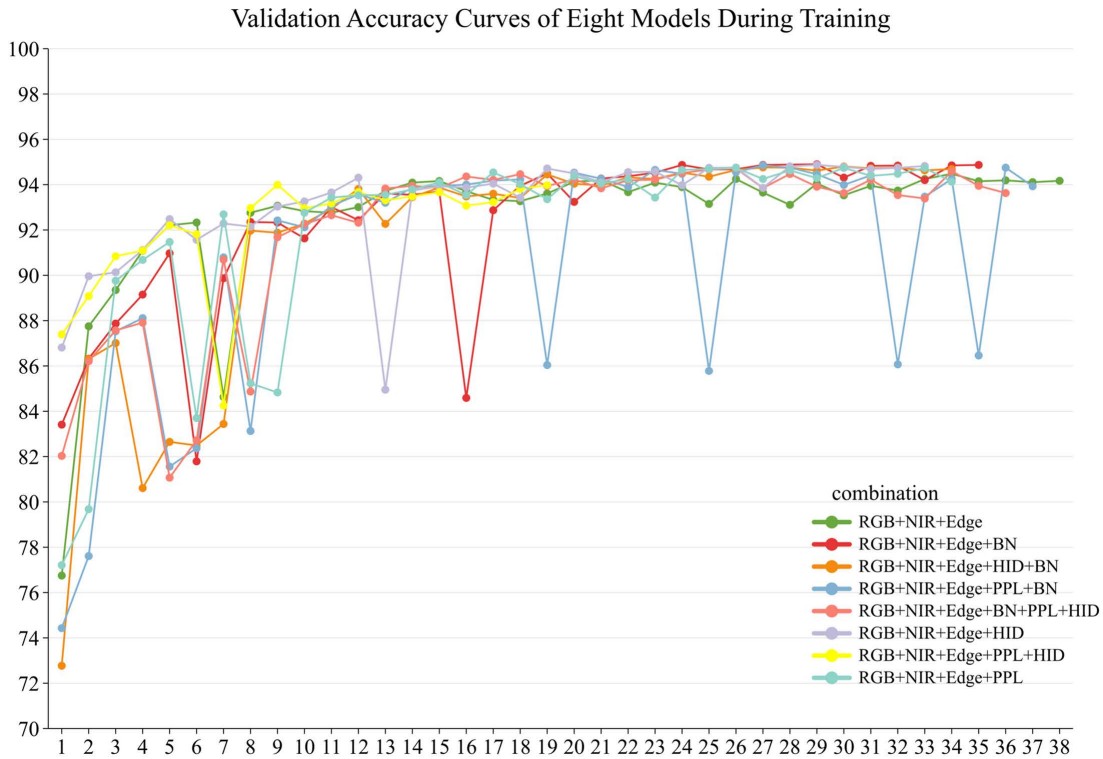

Fig 7. Comparative Validation Accuracy Curves of the Eight Models.

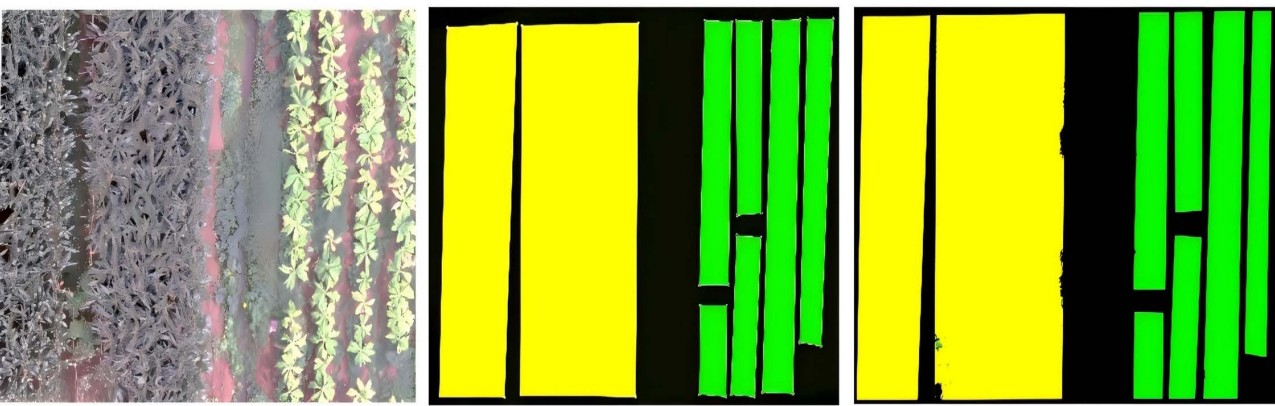

Fig 8. ResNet50_BN model classification outcomes.

higher than that of the other models. This signifies considerable instability during training, either owing to structural redundancy or inadequate coupling between components, resulting in optimization challenges.

The incorporation of the BN layer significantly improves the training stability and generalization capacity of the model. The integration of hidden layers and pyramid pooling layers necessitates a thorough evaluation of various parameters to prevent performance decline owing to excessive structural complexity. Subsequent studies should investigate complementary mechanisms and fusion procedures among diverse components to achieve high-precision multimodal image classification.

Fig 8 UAV imagery from Prediction Zone. Data availability: The UAV imagery presented in this figure is the original work of the authors. The underlying image data are available as part of the complete dataset in the Zenodo repository: https://doi.org/10.5281/zenodo.17451761.

## Discussion

Band fusion tests revealed that the full-spectrum combinations surpassed the pure visible-light setup. Performance improvement arises not from mere information aggregation but from the supplementary discriminative information offered by multispectral channels. Crops such as maize and flue-cured tobacco are posited to possess inherently different reflectance spectral properties in the infrared and red-edge bands, particularly for biophysical attributes such as vegetation growth status, leaf morphology, and chlorophyll concentration. The red-edge band has increased sensitivity to fluctuations in chlorophyll content, catching intricate features of visible light bands. Thus, multispectral channel fusion significantly enhances the dimensionality of the feature space, allowing deep learning models to utilize these inherent spectral variations. This method generates more distinct feature representations and alleviates ambiguity across spectrally similar object categories.

The enhanced experimental findings demonstrate that the BN layer effectively mitigates internal covariate shift, accelerates the training process, improves the convergence rate and generalization capability, and contributes to model stability, thus facilitating faster convergence during training. Conversely, while the pyramid pooling layer (PPL) is theoretically capable of increasing the model's proficiency in recognizing multiscale objects by assimilating multiscale information, its impact on enhancing classification accuracy is negligible in this investigation. This study suggests that PPL may not completely leverage its theoretical benefits, likely because of the simplicity of crop types and scale distributions in the study area, resulting in underutilization of the multiscale information captured by PPL in the classification task. Furthermore, the incorporation of PPL escalates the model's computational complexity, thereby prolonging the training time, whereas the improvement in classification accuracy remains negligible.

The inclusion of hidden layers enhances the model's feature extraction capabilities but also increases its complexity, potentially leading to overfitting and reducing the model's generalization ability. This study found that the addition of a hidden layer improved the model's performance on the training set but reduced its performance on the validation set. This suggests that the incorporation of the hidden layer causes the model to overfit the training data, resulting in poor generalization to new data. Furthermore, the addition of the hidden layer increases the model's parameter count, complicating both the training and inference processes, which consequently raises the computational cost.

In addition to classification accuracy, computing efficiency is a crucial criterion for assessing the practical applicability of a model in UAV remote sensing systems. In this study, we performed a meticulous assessment of this issue. The results demonstrate that although the batch normalization layer incurs a minor computational cost, it improves the overall training efficiency by stabilizing the training process and expediting convergence. This is advantageous for model deployment in resource-limited environments. Despite the pyramid pooling module and supplementary hidden layers providing minimal performance improvements in the analyzed scenarios, this predominantly indicates the data attributes and size constraints of the present study. Importantly, these modules did not result in substantial increases in computing overhead, and their potential across diverse complexity levels of agricultural remote-sensing tasks merits further investigation. The resulting architecture demonstrates a deliberate compromise between theoretical performance and engineering practicality, providing insights for analogous applications: guaranteeing training robustness via foundational optimizations, such as BN, is more pragmatic than indiscriminately augmenting complexity. Future lightweight architectures and model compression methods represent viable approaches for attaining optimal inference efficiency.

## Conclusion

This study focused on Yusho Town, Chengjiang City, Yuxi City, Yunnan Province, utilizing an unmanned aerial vehicle (UAV) equipped with a multispectral sensor to collect data across five bands. The multiband data was then input into

an enhanced ResNet50 residual network model to improve the classification accuracy of roasted tobacco and maize. By comparing the results of the ablation and improvement experiments and evaluating the accuracy, the optimal band combinations for classifying roasted tobacco and maize, along with the model enhancement strategy, were identified. This provides data support for optimizing the model's performance and structure, while also offering technical assistance for the precise classification of roasted tobacco and maize on a small scale. The following conclusions can be drawn from this study:

The inclusion of the infrared band significantly improves the classification accuracy of roasted tobacco and maize, whereas the addition of the red-edge band alone does not substantially enhance the classification accuracy of these crops and may even reduce it. The optimal classification precision was attained with the simultaneous incorporation of the infrared and red-edge bands, resulting in a 0.93% enhancement in classification precision, a 0.93% improvement in recall, a 1.4% increase in the kappa index, and a 0.92% increase in the F1 score relative to the prior condition without the bands.

The ResNet50 model demonstrates superior efficacy in the multispectral image classification of roasted tobacco and maize when five bands are used as model inputs, achieving classification precision, recall, and F1 score exceeding 93%, along with a Kappa index of 92%.

Utilizing the ResNet50 model, various combinations of the BN layer, pyramid pooling layer, and hidden layer were examined. Excluding the simultaneous incorporation of the pyramid pooling layer and hidden layer, the remaining six combinations enhanced the classification accuracy. Notably, the introduction of the BN layer alone resulted in the most significant improvements, with accuracy increasing by 0.42%, precision by 0.51%, recall by 0.42%, the Kappa index by 0.63%, and the F1 score by 0.44%.

The incorporation of the batch normalization layer, pyramid pooling layer, and hidden layer is expected to increase the classification accuracy; however, it simultaneously increases the model complexity and training duration. Furthermore, the efficacy of these components is closely tied to the characteristics of the dataset, including its quantity, type, and distribution, suggesting that greater complexity does not necessarily lead to superior results.

## Acknowledgments

First and foremost, I extend my heartfelt gratitude to my tutor, Professor Jianxiong Wang (J.X. W), whose invaluable guidance and financial support were crucial to the completion of this thesis. Additionally, I deeply appreciate my classmate Shixian Lu (S.X. L) and my two close friends, who offered insightful suggestions and hands-on assistance during the experimental and writing phases.

## Author contributions

**Conceptualization:** Xiang Feng.

**Data curation:** Chenwei Xu, Xiang Feng.

**Formal analysis:** Chenwei Xu, Jianxiong Wang.

**Funding acquisition:** Jianxiong Wang.

**Investigation:** Chenwei Xu, Shixian Lua.

**Methodology:** Chenwei Xu, Xiang Feng, Jianxiong Wang.

**Project administration:** Chenwei Xu, Xuelin Zhang, Jianxiong Wang.

**Resources:** Xuelin Zhang.

**Supervision:** Chenwei Xu, Shixian Lua, Xuelin Zhang, Jianxiong Wang.

**Validation:** Chenwei Xu, Xiang Feng, Jianxiong Wang.

**Visualization:** Chenwei Xu.

**Writing – original draft:** Chenwei Xu.

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
