## [Decision Letter · Decision Letter 0]

24 Sep 2025

Dear Dr. Xu,

Thank you for submitting your manuscript to PLOS ONE. After careful consideration, we feel that it has merit but does not fully meet PLOS ONE’s publication criteria as it currently stands. Therefore, we invite you to submit a revised version of the manuscript that addresses the points raised during the review process.

We look forward to receiving your revised manuscript.

Kind regards,

Muhammad Shahid Farid, Ph.D.

Academic Editor

PLOS ONE

Journal Requirements:

4. In the online submission form, you indicated that the data that support the findings of this study are available on request from the corresponding author. The data are not publicly available due to privacy or ethical restrictions.

This work was supported by the National Key R & D Program (2024YFD1700104).

First and foremost, I extend my heartfelt gratitude to my tutor, Professor Jianxiong Wang, whose invaluable guidance and financial support were crucial to the completion of this thesis. Additionally, I deeply appreciate my classmate Shixian Lu and my two close friends, who offered insightful suggestions and hands-on assistance during the experimental and writing phases. This work was supported by the National Key R & D Program (2024YFD1700104).

This work was supported by the National Key R & D Program (2024YFD1700104).

7. Please amend your list of authors on the manuscript to ensure that each author is linked to an affiliation. Authors’ affiliations should reflect the institution where the work was done (if authors moved subsequently, you can also list the new affiliation stating “current affiliation:….” as necessary).

8. We note that Figure(s) 1, 2 and 6, in your submission contain [map/satellite] images which may be copyrighted. All PLOS content is published under the Creative Commons Attribution License (CC BY 4.0), which means that the manuscript, images, and Supporting Information files will be freely available online, and any third party is permitted to access, download, copy, distribute, and use these materials in any way, even commercially, with proper attribution. For these reasons, we cannot publish previously copyrighted maps or satellite images created using proprietary data, such as Google software (Google Maps, Street View, and Earth). For more information, see our copyright guidelines: http://journals.plos.org/plosone/s/licenses-and-copyright.

a. You may seek permission from the original copyright holder of Figure(s) 1, 2 and 6, to publish the content specifically under the CC BY 4.0 license.

Reviewers' comments:

Reviewer's Responses to Questions

**Comments to the Author**

1. Is the manuscript technically sound, and do the data support the conclusions?

Reviewer #1: Yes

Reviewer #2: Yes

2. Has the statistical analysis been performed appropriately and rigorously?

Reviewer #1: Yes

Reviewer #2: I Don't Know

3. Have the authors made all data underlying the findings in their manuscript fully available?

Reviewer #1: No

Reviewer #2: Yes

4. Is the manuscript presented in an intelligible fashion and written in standard English?

Reviewer #1: Yes

Reviewer #2: Yes

Reviewer #1: 1. The enhancements to ResNet50 (BN, PPL, hidden layers) are incremental. The manuscript should better articulate the novelty compared with prior deep learning approaches for UAV-based crop classification (e.g., DeepLab V3+, Bi-LSTM-Attention, MSNet). A clearer positioning of the contribution in relation to state-of-the-art models would strengthen the work.

2. The ablation study indicates RGB+NIR+Edge as optimal, but the physiological/biophysical reasoning behind this combination is underexplored. Including vegetation index correlations (NDVI, CCI, etc.) or spectral signatures of maize and tobacco would justify why NIR and Edge bands are most discriminative.

3. Several equations (e.g., (1)–(4) for ResNet layer transformations, (5) for PPL, and (6)–(12) for BN propagation) are presented in a very generic form without proper derivation or sufficient explanation of the variables (e.g., fl,Wl,xl). To improve clarity, the authors should explicitly define all symbols at first use and connect the equations to their role in the modified ResNet50 architecture.

4. The mathematical formulations for BN and hidden layers (e.g., (6)–(14)) are largely restatements of standard definitions from deep learning literature. The manuscript would be stronger if these equations were contextualized to this specific application (UAV multispectral crop classification). For example, authors could explain how batch normalization interacts with multispectral input channels or why certain placements of BN (after convolution vs. hidden layer) theoretically affect convergence in their model.

5. The dataset is collected from a single location and two dates (Aug 19, 27, 2024). This may limit generalizability. The authors should discuss dataset diversity (seasonal variation, crop growth stages, environmental conditions) and whether the trained model would transfer to other geographies or times.

6. While preprocessing steps (radiometric calibration, mosaicking) are described, details about class imbalance handling, labeler expertise, and annotation quality control are missing. Since mislabeling can strongly affect results, annotation reliability should be clarified.

7. Results are compared only across enhanced ResNet50 variants. To demonstrate true advancement, comparisons with other architectures (e.g., EfficientNet, DenseNet, Vision Transformers) or traditional machine learning methods (e.g., SVM, RF with vegetation indices) should be included.

8. The discussion notes that PPL and hidden layers increased complexity without meaningful gains. A quantitative analysis of training time, parameter counts, FLOPs, or inference latency would make this trade-off clearer, especially since UAV systems often need efficient on-board or near-real-time processing.

9. Only overall accuracy, precision, recall, kappa, and F1 are reported. Given the class imbalance (tobacco vs. maize vs. background), class-wise metrics and ROC/PR curves would provide deeper insights into performance.

10. The study mentions overfitting with hidden layers, but no learning curves are provided. Including training/validation loss curves would help verify convergence behavior and overfitting tendencies.

11. Page 13: The manuscript mentions 5-fold cross-validation but reports only mean values. Reporting standard deviations would better indicate robustness and variance across folds.

Reviewer #2: The presented work in this manuscript is technically vigorous and well-structured, with strong experimental validation, the paper is suitable for publication. However, it is suggested for the authors to include a brief runtime/complexity analysis or at least a qualitative discussion of computational feasibility..

**Do you want your identity to be public for this peer review?** For information about this choice, including consent withdrawal, please see our Privacy Policy

Reviewer #1: No

Reviewer #2: No

---

## [Author Response · Author response to Decision Letter 1]

5 Nov 2025

Dear Editors and Reviewers,

Thank you for your valuable comments and the opportunity to revise our manuscript. We have thoroughly addressed all points raised in the decision letter within our detailed point-by-point response, which has been uploaded as a separate file titled "Response to Reviewers".

In the attached document, we have provided a comprehensive reply to each comment, detailing the specific changes made to the manuscript. We believe these revisions have significantly strengthened our work, and we are grateful for the insightful feedback.

Sincerely,

Jianxiong Wang

Email: wangjx@ynau.edu.cn

---

## [Decision Letter · Decision Letter 1]

9 Dec 2025

Crop Classification with UAV Multispectral Remote Sensing, Employing an Enhanced ResNet50 Residual Network

PONE-D-25-38216R1

Dear Dr. Xu,

We’re pleased to inform you that your manuscript has been judged scientifically suitable for publication and will be formally accepted for publication once it meets all outstanding technical requirements.

Kind regards,

Muhammad Shahid Farid, Ph.D.

Academic Editor

PLOS One

Additional Editor Comments (optional):

Reviewers' comments:

Reviewer's Responses to Questions

**Comments to the Author**

Reviewer #1: All comments have been addressed

Reviewer #2: All comments have been addressed

2. Is the manuscript technically sound, and do the data support the conclusions?

Reviewer #1: Yes

Reviewer #2: Yes

3. Has the statistical analysis been performed appropriately and rigorously?

Reviewer #1: Yes

Reviewer #2: I Don't Know

4. Have the authors made all data underlying the findings in their manuscript fully available?

Reviewer #1: Yes

Reviewer #2: Yes

5. Is the manuscript presented in an intelligible fashion and written in standard English?

Reviewer #1: Yes

Reviewer #2: Yes

Reviewer #1: Accept the manuscript in the current form. Authors addressed all of the suggested comments satisfactorily.

Reviewer #2: (No Response)

**Do you want your identity to be public for this peer review?** For information about this choice, including consent withdrawal, please see our Privacy Policy

Reviewer #1: No

Reviewer #2: No

---

## [Editor Report · Acceptance letter]

PONE-D-25-38216R1

PLOS One

Dear Dr. Xu,

I'm pleased to inform you that your manuscript has been deemed suitable for publication in PLOS One. Congratulations! Your manuscript is now being handed over to our production team.

Kind regards,

on behalf of

Dr. Muhammad Shahid Farid

Academic Editor

PLOS One